# Unveiling the Role of Graphite Morphology in Ductile Iron: A 3D FEM-Based Micromechanical Framework for Damage Evolution and Mechanical Performance Prediction with Applicability to Multiphase Alloys

**DOI:** 10.3390/ma18225128

**Published:** 2025-11-11

**Authors:** Jing Tao, Yufei Jiang, Shuhui Xie, Yujian Wang, Ziyue Zhou, Lingxiao Fu, Chengrong Mao, Lingyu Li, Junrui Huang, Shichao Liu

**Affiliations:** 1Hubei Key Laboratory of Intelligent Transportation Technology and Device, School of Mechanical and Electrical Engineering, Hubei Polytechnic University, Huangshi 435003, China; jingtao_hbpu@126.com (J.T.); yfjiang_1991@126.com (Y.J.); xie406050@163.com (S.X.);; 2Riyue Research Institute, Riyue Heavy Industry Co., Ltd., Ningbo 315000, China; 3School of Mechanical Engineering, Dalian University of Technology, Dalian 116000, China; 4School of Iron and Steel, Soochow University, Suzhou 215021, China; sc_liu@suda.edu.cn

**Keywords:** ductile iron, gray iron, graphite morphology, finite element method (FEM), fracture mechanism

## Abstract

The mechanical performance of cast iron is strongly governed by the morphology of its graphite phase, yet establishing a quantitative link between microstructure and macroscopic properties remains a challenge. In this study, a three-dimensional finite element method (FEM)-based micromechanical framework is proposed to analyze and predict the mechanical behavior of cast iron with representative graphite morphologies, spheroidal and flake graphite. Realistic representative volume elements (RVEs) are reconstructed based on experimental microstructural characterization and literature-based X-ray computed tomography data, ensuring geometric fidelity and statistical representativeness. Cohesive zone modeling (CZM) is implemented at the graphite/matrix interface and within the graphite phase to simulate interfacial debonding and brittle fracture, respectively. Full-field simulations of plastic strain and stress evolution under uniaxial tensile loading reveal that spheroidal graphite promotes uniform deformation, delayed damage initiation, and enhanced ductility through effective stress distribution and progressive plastic flow. In contrast, flake graphite induces severe stress concentration at sharp tips, leading to early microcrack nucleation and rapid crack propagation along the flake planes, resulting in brittle-like failure. The simulated stress–strain responses and failure modes are consistent with experimental observations, validating the predictive capability of the model. This work establishes a microstructure–property relationship in multiphase alloys through a physics-informed computational approach, demonstrating the potential of FEM-based modeling as a powerful tool for performance prediction and microstructure-guided design of cast iron and other heterogeneous materials.

## 1. Introduction

The development of metallic materials has long relied on empirical, trial-and-error approaches, which are time-consuming, costly, and often lack mechanistic insight [1,2]. Despite significant progress in alloy design, the relationship between microstructure and mechanical performance, especially fracture behavior, is still predominantly explored through experimental characterization [3]. While techniques such as electron microscopy and mechanical testing provide valuable data, they offer only snapshots of material behavior and limited access to the dynamic evolution of internal deformation and damage mechanisms, particularly under in situ loading conditions.

A fundamental challenge arises from the inherent opacity of metals, which prevents direct observation of internal microstructural responses during plastic deformation. For multiphase alloys with complex microstructures, such as those containing dispersed second-phase particles, lamellar or nodular graphite, or heterogeneous phase distributions, researchers are typically restricted to post-mortem analysis of fracture surfaces to infer failure mechanisms [4,5,6,7]. This indirect approach introduces a high degree of uncertainty and subjectivity, making it difficult to establish causal links between microstructural features and macroscopic properties. Consequently, there remains a critical gap in establishing a predictive, physics-based framework that can quantitatively link microstructural morphology to mechanical performance [8]. The absence of such a model hinders the rational design of new materials and limits the ability to optimize existing ones. To overcome these limitations, there is a pressing need for advanced computational strategies that can simulate the three-dimensional microstructural evolution and local stress–strain fields during deformation, thereby enabling a deeper, more systematic understanding of fracture mechanisms in complex alloy systems.

In recent years, with significant advances in computational power and multiscale modeling techniques, the finite element method (FEM) has emerged as a powerful tool for investigating the micromechanical behavior of material [9,10,11]. FEM enables accurate simulation of local stress and strain fields, as well as damage evolution, based on realistic or reconstructed three-dimensional microstructural models. This capability overcomes the limitations of experimental methods in terms of observation depth and spatiotemporal resolution. Particularly for opaque metallic materials, FEM provides a "virtual experiment" approach, allowing researchers to visualize and quantify internal deformation heterogeneity, stress concentration zones, and crack initiation and propagation paths, without relying on direct physical observation.

In the field of multiphase alloys, FEM has been widely applied to analyze interfacial behavior between secondary phase particles and the matrix, dislocation pile-up effects, development of localized plastic zones, and microcrack formation mechanisms [12,13,14,15]. For instance, researchers have used FEM to reveal how stress concentrations around hard particles promote crack initiation, and to elucidate the role of microscale interfaces, such as grain and phase boundaries, in crack deflection or bridging. Furthermore, by integrating experimentally obtained microstructural data (e.g., from Electron Backscatter Diffraction (EBSD) or X-ray CT) to construct realistic models [16,17,18], FEM has significantly enhanced the physical fidelity and predictive accuracy of simulations.

Despite these advances, most existing simulations are still limited to two-dimensional simplifications or idealized geometric configurations, which fail to capture the complex topological relationships of microstructures in three dimensions and their influence on deformation compatibility and fracture trajectories [19]. Especially for alloy systems with pronounced microstructural anisotropy or complex phase distributions, such as the flake graphite network in gray iron or the nodular graphite array in ductile iron, three-dimensional FEM modeling not only enables a more accurate description of stress transfer and localized strain mismatch but also provides a critical bridge for establishing quantitative relationships between "microstructure, local field, and macroscopic properties".

Therefore, the development of 3D finite element models based on realistic microstructures offers not only deeper insights into fracture mechanisms in multiphase alloys but also a viable pathway toward building a generalizable framework for material property prediction. In this study, ductile iron and gray iron are selected as representative multiphase alloy systems. High-fidelity 3D FEM models are constructed to systematically investigate the influence of different graphite morphologies on matrix deformation behavior and crack propagation paths. Furthermore, the general applicability and broader value of this modeling strategy for a wide range of multiphase alloys are discussed.

In this work, a high-fidelity 3D finite element modeling framework is developed to investigate the microstructure–property relationships in multiphase alloys, using ductile iron and gray iron as representative systems. Realistic microstructural models incorporating nodular and flake graphite morphologies are reconstructed based on experimental characterization, enabling full-field simulations of plastic deformation and crack propagation under tensile loading. The evolution of local stress concentrations, strain localization, and damage initiation is analyzed in three dimensions, revealing the critical role of graphite morphology in governing fracture mechanisms. This study presents three key innovations: (1) the construction of physically accurate 3D microstructural models that fully capture the complex topology and spatial distribution of second-phase graphite in cast irons; (2) the implementation of a robust simulation strategy capable of tracking progressive damage and crack evolution in heterogeneous microstructures, bridging microscale deformation with macroscale failure; and (3) the demonstration of a generalizable modeling approach that can be extended to a wide range of multiphase alloys beyond cast irons, paving the way for predictive microstructure-based material design. This work thus establishes a paradigm shift from empirical trial-and-error methods toward physics-informed, computational-driven materials development.

## 2. Experimental

### 2.1. Material Preparation

This study is based on laboratory-scale experiments conducted in collaboration with Riyue Heavy Industry Co., Ltd. (Ningbo, China), where ductile iron (QT400-18AL) samples were produced under industrial-relevant conditions. The samples mainly contains C (3.4–3.8%), Si (1.8–2.1%), Mg (0.04–0.06%), P (0.04%) and S (0.02%).

To gain deeper insight into the formation mechanisms of graphite distortion, laboratory-scale castings were performed using two ductile iron alloy variants. The first alloy had a nominal composition of Fe-3.6C-2.0Si (in wt.%), while the second included an addition of 0.04 wt.% Mg (Fe-3.6C-2.0Si-0.04Mg), with compositions adjusted using appropriate ferroalloys. These formulations were designed to match those used in industrial production, ensuring the practical relevance of the experimental findings. By controlling the magnesium content, different graphite morphologies were obtained, enabling a systematic investigation of how alloying conditions influence graphite deformation and the associated microstructural evolution under realistic casting conditions.

### 2.2. Microstructure Characterization and Tensile Test

Samples from both industrial production runs and laboratory castings were collected for microstructural analysis. Field emission scanning electron microscopy (SEM, Zeiss Supra 55, Carl Zeiss Microscopy GmbH, Jena, Germany) combined with energy-dispersive X-ray spectroscopy (EDS, Carl Zeiss Microscopy GmbH, Jena, Germany) was employed for high-resolution imaging and elemental characterization, including point analyses and area mapping. These techniques provided detailed information on phase distribution, local chemistry, and graphite morphology, facilitating a comprehensive evaluation of the relationship between microstructure and mechanical performance in ductile iron.

Tensile testing was carried out to assess the mechanical behavior of the ductile iron, with particular emphasis on its stress–strain response under uniaxial loading. Tests were conducted on a Sans universal testing machine (Shenzhen Suns Technology Co., Ltd., Shenzhen, China.) at a constant crosshead speed of 3 mm/min. Strain was measured using a video extensometer to ensure high accuracy and non-contact deformation monitoring. To guarantee statistical reliability, each condition was tested with at least three replicate samples, ensuring consistent and reproducible mechanical data. The concrete process is illustrated by Figure 1. The tensile tests were conducted in accordance with the Chinese national standard GB/T 228.1-2021 [20].

### 2.3. FEM Simulation

For the finite element analysis (FEM), numerical simulations were performed using the commercial software Abaqus. A 3D stress model was adopted to represent the specimen cross-section, which is appropriate for thin geometries under in-plane tensile loads. To capture the initiation and propagation of cracks, cohesive zone modeling (CZM) was implemented by inserting cohesive elements along potential fracture paths. This approach enables a physically based representation of the fracture process, allowing for detailed analysis of how graphite morphology and interfacial degradation influence local stress fields and overall failure mechanisms. The complete set of material constitutive models employed in the FEM simulations is detailed in Reference [19]. The parameters used in FEM for modeling have been listed in Table 1.

To capture the initiation and propagation of cracks, cohesive zone modeling (CZM) was implemented by inserting cohesive elements along potential fracture paths, specifically at the graphite/matrix interface and within the graphite phase to simulate interfacial debonding and intra-particle fracture, respectively.

A traction–separation law (TSL) was adopted to characterize the cohesive behavior, which defines the relationship between the cohesive tractions (normal and shear) and the relative displacement (separation) across the cohesive surfaces. The damage evolution follows a bilinear or exponential softening law, depending on the material system and calibration. The concrete parameters used to characterize the CZM element originate from the previous work [19].

## 3. Results and Discussion

### 3.1. Reconstruction of Realistic 3D Microstructural Models for Ductile and Gray Iron

The microstructural characteristics of the two iron alloys were first examined through optical microscopy across multiple cross-sectional planes to assess the representativeness and homogeneity of the microstructure. As shown in Figure 2, the ductile iron samples (Figure 2(a_1_–a_3_)) exhibit a typical microstructure composed of a ferritic matrix with uniformly distributed globular graphite nodules. These nodules are spherical or near-spherical in shape and vary slightly in size, but their overall morphology remains consistent across the three observed planes, indicating a relatively homogeneous distribution throughout the casting. This uniformity is crucial for ensuring reliable mechanical performance and for constructing accurate representative volume elements (RVEs) [21] in subsequent finite element simulations.

In contrast, the gray iron samples Figure 2(b_1_–b_3_) display a markedly different microstructure characterized by flake graphite dispersed within the ferritic matrix. The graphite flakes appear elongated and interconnected, forming a network that disrupts the continuity of the metallic matrix. Notably, the morphology and orientation of these flakes show significant variation between different planes, suggesting a degree of anisotropy in the microstructure due to directional solidification or other processing factors. This inherent heterogeneity and anisotropy pose greater challenges for modeling and predicting the mechanical behavior compared to ductile iron.

The distinct morphologies of the second-phase graphite, with globular graphite in ductile iron and flake graphite in gray iron, are directly linked to their differing mechanical responses. The globular graphite acts as a more favorable stress concentrator, promoting plastic deformation of the surrounding matrix and enabling higher ductility. In contrast, the sharp edges and high aspect ratio of the flake graphite serve as potent sites for crack initiation under applied load, leading to brittle fracture behavior. The detailed characterization presented in Figure 2 provides essential input data for building realistic three-dimensional microstructural models, which are subsequently used to simulate the plastic deformation and fracture evolution in these multiphase systems.

To construct physically representative 3D microstructural models for finite element simulation, the morphological characteristics of the second-phase graphite were derived from real industrial X-ray computed tomography (CT) results reported in the literature Reference [22]. These CT scans provide valuable insights into the complex three-dimensional architecture of flake graphite in gray iron and nodular graphite in ductile iron, which cannot be fully captured by conventional 2D metallography.

Based on these reference morphologies, a morphology-informed 3D reconstruction algorithm was employed to generate RVEs that emulate the realistic spatial distribution and geometric complexity of graphite. The microstructural parameters used in the modeling, particularly the volume fraction of graphite, were calibrated to match the statistical results obtained from quantitative metallographic analysis of the actual samples in this study (Figure 3a,d).

As shown in Figure 3b,e, the reconstructed 3D graphite phases exhibit morphologies closely resembling those observed in real cast iron: an interconnected network of high-aspect-ratio flakes for gray iron and a dispersion of near-spherical nodules for ductile iron. The corresponding matrix structures (Figure 3c,f) were generated by removing the graphite phase, resulting in voids that preserve the original spatial configuration and interfacial geometry.

This hybrid approach combines realistic morphology from literature CT data with experimentally determined microstructural parameters (e.g., volume fraction), and ensures that the simulation models are both geometrically faithful and statistically representative. It enables a more accurate investigation of the microstructure–property relationships in multiphase alloys, even in the absence of direct 3D imaging of the specific samples.

Although the RVE does not explicitly resolve the complete eutectic cell boundaries, the reconstructed flake network preserves the local morphology, orientation correlation, and interconnectivity observed in high-fidelity X-ray CT data [22], which are critical features governing crack initiation and propagation in gray iron. This level of representation is considered sufficient for the comparative micromechanical analysis of morphology-dependent mechanical response pursued in this study.

It should be noted that the simulated RVE assumes a clean ferritic matrix representative of high-purity laboratory-scale material [19], rather than industrial-grade cast iron containing impurities and inclusions that severely limit ductility. Therefore, the absolute ductility values are not intended to match industrial data, but the model is suitable for comparative analysis of morphology-dependent deformation mechanisms.

### 3.2. Full-Field Simulation of Plastic Deformation and Fracture Evolution in Multiphase Alloys

The microstructure of the fractured surface was further analyzed using energy-dispersive X-ray spectroscopy (EDS), as shown in Figure 4. The SEM image (Figure 4a) reveals several key features associated with the failure mechanism of ductile iron under tensile loading. Notably, some graphite particles (GPs) exhibit fracture, indicating that they have undergone brittle cleavage during deformation. In addition, clear interfacial debonding is observed between the graphite and the surrounding ferritic matrix, suggesting a weak bonding at the interface. Furthermore, certain GPs appear to have undergone plastic movement or displacement, consistent with the idea that these particles can act as stress concentrators and promote localized plastic flow.

The corresponding EDS elemental maps (Figure 4b–e) provide complementary chemical information. The Fe map (Figure 4b) highlights the metallic matrix phase, while the C map (Figure 4c) clearly outlines the graphite particles due to their high carbon content. The Si map (Figure 4d) shows a uniform distribution in the matrix, typical of ferritic ductile iron. The distribution of O is shown in Figure 4e. 

Together, these results suggest that the fracture process in ductile iron involves a combination of graphite particle fracture, interfacial debonding, and matrix plasticity. The observed debonding and oxygen accumulation at the interface imply that the mechanical integrity of the GP/matrix interface plays a critical role in determining the overall ductility and strength of the material.

Therefore, in this study, a cohesive zone model (CZM) is incorporated at the graphite/matrix interface to explicitly capture the interfacial debonding behavior [19,21,22,23]. The cohesive zone is characterized by a traction–separation law that governs the evolution of interfacial damage under normal and shear loading conditions. Additionally, the brittle fracture of the graphite particles themselves is accounted for by assigning appropriate fracture criteria to the bulk graphite phase. This combined approach enables a more realistic simulation of the multiscale failure mechanisms in ductile iron, including crack initiation at the interface, progressive debonding, and particle fracture.

Figure 5 illustrates the progressive damage evolution in a RVE of gray iron under uniaxial tensile loading, simulated using a multiscale finite element model incorporating cohesive zone elements (COH3D6) at both the graphite/matrix interface and within the flake graphite particles. The simulation captures the full deformation history from ε = 0 to ε = 22.5%, highlighting three key aspects: overall microstructural response, interfacial debonding behavior, and intra-particle fracture.

The top row Figure 5(a_1_–a_4_) presents the whole model, showing the α-Fe matrix (green), graphite particles (gray), and cohesive zones (red). As strain increases, initial plastic deformation occurs in the ferritic matrix, while cracks begin to nucleate at the interfaces and propagate along preferred paths. By ε = 22.5%, extensive crack networks are formed, indicating macroscopic failure initiation.

The middle row Figure 5(b_1_–b_4_) focuses on the graphite/matrix interface, where cohesive zones with zero thickness are inserted between the phases. With increasing strain, these cohesive elements undergo traction separation and eventually fail via element deletion, representing interfacial debonding. This process is clearly visualized as red regions disappearing or fragmenting, particularly evident at ε = 15% and 22.5%, confirming that the interface acts as a primary site for damage initiation.

The bottom row Figure 5(c_1_–c_4_) zooms into the flake structure of individual graphite particles, revealing their internal morphology. Initially intact, the graphite begins to fracture at higher strains due to high local stress concentrations. The simulation models this brittle fracture by deleting cohesive elements within the graphite flakes, resulting in visible crack propagation and fragmentation-consistent with experimental observations of graphite cleavage.

Figure 6 presents a comprehensive numerical simulation of damage evolution in ductile iron containing spherical graphite particles, subjected to uniaxial tensile loading. The model employs a three-dimensional RVE with COH3D6 inserted at both the graphite/matrix interface and within the graphite phase to simulate interfacial debonding and intra-particle fracture.

The top row Figure 6(a_1_–a_4_) shows the whole microstructure, where the green mesh represents the α-Fe ferritic matrix (C3D4), gray regions are spherical graphite particles (C3D4), and red areas denote cohesive zones (COH3D6). As strain increases from 0 to 22.5%, plastic deformation accumulates in the matrix, while cracks nucleate at the interfaces and propagate through the matrix. At ε = 22.5%, significant void coalescence and macroscopic cracking are observed, indicating the onset of final failure.

The middle row Figure 6(b_1_–b_4_) focuses on the graphite/matrix interface, highlighting the evolution of cohesive zones. Initially intact, these zero-thickness cohesive elements begin to fail progressively as stress builds up. By ε = 15%, localized debonding is evident, and at ε = 22.5%, extensive element deletion occurs, representing interfacial separation. This process weakens load transfer and promotes crack growth into the matrix.

The bottom row Figure 6(c_1_–c_4_) zooms into the individual spherical graphite particles, revealing their internal structure and fracture behavior. Although spherical graphite is generally considered more favorable for ductility due to reduced stress concentration, the simulation shows that under high strain, brittle fracture still occurs within the graphite itself, especially near the equator or along preferred crystallographic planes. The appearance of red cohesive zones and fragmented particles indicates crack initiation and propagation inside the graphite, consistent with experimental observations of microcracking in graphite nodules.

A direct comparison between Figure 5 (flake graphite) and Figure 6 (globular graphite) highlights the significant influence of graphite morphology on the damage evolution in cast iron. In the case of flake graphite, the high aspect ratio and sharp edges lead to severe stress concentration, resulting in early interfacial debonding and rapid crack propagation along the flake boundaries, even at low strain levels. In contrast, the spherical morphology of graphite in ductile iron effectively mitigates stress concentration, delaying both interfacial failure and intra-particle fracture. As a result, the globular graphite structure exhibits a more gradual damage progression and enhanced ductility. However, both morphologies ultimately undergo graphite/matrix debonding and internal brittle fracture at high strains, indicating that while spheroidization improves mechanical performance, it does not eliminate the inherent brittleness of the graphite phase. This comparative analysis underscores the critical role of microstructure design in optimizing the strength–ductility balance in cast irons.

The simulated stress–strain responses of cast iron with different graphite morphologies are presented in Figure 7. The curve for material containing globular graphite exhibits a higher yield strength, a peak tensile strength of approximately 450 MPa, and significant plastic deformation up to about 15% strain before fracture. In contrast, the response of material with flake graphite shows markedly lower ductility, with an ultimate strength of around 250 MP. The early softening behavior observed in the flake graphite case is attributed to rapid crack propagation along the flake interfaces, which act as preferential paths for fracture.

While the absolute stress and strain values may vary slightly from experimental measurements, the overall trend of the simulated curves is in good agreement with previously reported results [19]. The superior strength and ductility of the globular graphite structure, as well as the brittle nature of the flake graphite morphology, are consistent with established experimental observations. The simulation successfully captures the characteristic mechanical behavior associated with each microstructure, including the onset of yielding, strain hardening, and post-peak softening.

This alignment in trend, particularly the clear distinction in damage evolution and load-bearing capacity between the two morphologies, demonstrates that the modeling approach adequately represents the underlying deformation and failure mechanisms. The results confirm that the implemented cohesive zone model can effectively reflect how microstructural features influence macroscopic mechanical performance, providing a reliable basis for predicting the mechanical response of cast iron under tensile loading.

### 3.3. Revealing the Influence Mechanism of Secondary Phase Morphology on Fracture Behavior

The evolution of plastic strain distribution within a RVE of ductile cast iron is illustrated in Figure 8, showing the progressive development of deformation from elastic to failure stages under uniaxial tension. At zero applied strain (Figure 8a), no plastic deformation is observed, with the entire matrix remaining in the elastic regime. As the strain increases to 3% (Figure 8b), localized plastic zones begin to emerge near the interfaces between the ferritic matrix and spherical graphite particles, indicating the onset of yielding in regions of high stress concentration. At 6% strain (Figure 8c), the plastic strain has further accumulated, particularly around the particle boundaries and in areas adjacent to clusters of graphite. The strain distribution becomes increasingly heterogeneous, with localized bands forming along potential shear paths. By 9% strain (Figure 8d), significant plastic flow is evident throughout the matrix, and the color scale indicates that some regions have reached plastic strains exceeding 0.30. Notably, the maximum plastic strain continues to concentrate near the graphite-matrix interfaces, suggesting that these interfaces are primary sites for deformation localization. As the strain reaches 12% (Figure 8e), macroscopic damage becomes apparent: visible cracks begin to nucleate at the edges of certain graphite particles, and the plastic strain contours show pronounced discontinuities. These features indicate the initiation of decohesion at the interface and the start of microvoid formation. At 15% strain (Figure 8f), the crack propagation is well advanced, with multiple interfacial debonding events and coalescence of microcracks leading to the formation of a through-thickness fracture path. The plastic strain distribution now shows large-scale localization along the developing crack plane, consistent with final failure.

This sequence demonstrates that the deformation mechanism in ductile cast iron proceeds through three distinct stages: (1) initial yielding at particle-matrix interfaces; (2) progressive strain accumulation and localization in the matrix; and (3) crack nucleation, growth, and coalescence driven by interfacial debonding. The results align with previous experimental observations of strain localization in ductile iron, where failure is preceded by extensive plastic deformation and gradual damage accumulation. Although the absolute values of strain may vary depending on material composition and boundary conditions, the overall pattern of plastic strain evolution is consistent with established mechanical behavior.

The simulation captures the essential physics of ductile fracture in cast iron, highlighting the critical role of graphite morphology in controlling deformation localization and damage progression. The ability to visualize strain fields at different loading stages provides valuable insight into the micromechanical processes underlying macroscopic failure, supporting the use of such models for predicting material performance under complex loading conditions.

The progression of plastic strain in gray cast iron, characterized by flake graphite, is illustrated in Figure 9, revealing a fundamentally different deformation mechanism compared to ductile iron with spheroidal graphite. At zero applied strain (Figure 9a), the material remains fully elastic, with no plastic deformation observed. However, even at 3% strain (Figure 9b), localized plastic zones emerge rapidly along the edges of graphite flakes, indicating that stress concentrations are significantly higher due to the sharp geometry of the flake morphology. By 6% strain (Figure 9c), plastic strain has accumulated extensively along the flake boundaries, forming continuous shear bands that propagate through the matrix. The strain localization is pronounced and aligned with the orientation of the graphite flakes, suggesting that these features act as internal crack-like defects that promote early yielding and damage initiation. In contrast to the gradual strain accumulation seen in spheroidal graphite structures, the presence of elongated flakes leads to rapid strain concentration and early matrix degradation. At 9% strain (Figure 9d), visible cracks begin to nucleate at the tips of graphite flakes, and the surrounding matrix shows significant plastic flow. The high strain gradients near the flake surfaces indicate substantial stress transfer from the matrix to the interface, leading to interfacial debonding and microvoid formation. By 12% strain (Figure 9e), multiple cracks have formed and coalesced into a dominant fracture path, resulting in a macroscopic failure zone. The plastic strain field is highly localized along this path, with little further deformation occurring in other regions. At 15% strain (Figure 9f), the specimen has undergone catastrophic failure, with a large through-thickness crack spanning the entire RVE. The surrounding matrix exhibits limited plasticity, and the overall deformation remains confined to narrow bands adjacent to the flake interfaces. This behavior reflects the inherent brittleness of gray cast iron, where the flake graphite acts not only as a stress concentrator but also as a direct pathway for crack propagation.

In comparison to the ductile iron case (Figure 8), the key difference lies in the role of graphite morphology in influencing deformation and damage evolution. While spherical graphite particles allow for uniform strain distribution and progressive plastic flow, flake graphite introduces discontinuities that disrupt load transfer and promote early fracture. The flake structure effectively "tears" the surrounding matrix, reducing the effective load-bearing area and accelerating failure. As a result, the material fails at much lower strains despite similar initial yield strength, highlighting the detrimental effect of flake morphology on ductility and fracture resistance.

This simulation captures the essential micromechanisms responsible for the poor ductility of gray cast iron, demonstrating that the shape and orientation of graphite inclusions play a critical role in determining the mechanical response. The results align with experimental observations of brittle fracture in gray iron, where failure occurs via crack propagation along flake interfaces rather than through extensive plastic deformation. The contrast between Figure 8 and Figure 9 underscores the importance of microstructure in tailoring the mechanical performance of cast iron alloys.

In Figure 10, the progression of effective stress within a representative volume element (RVE) of ductile cast iron is depicted under increasing tensile strain. This sequence offers insights into how stress redistributes and concentrates as deformation progresses.

At zero strain (Figure 10a), the RVE exhibits uniform low stress across the matrix, indicative of an entirely elastic state. As the material is subjected to a 3% strain (Figure 10b), localized regions of elevated stress begin to emerge around spherical graphite particles. These areas suggest initial yielding points where stress concentrations are heightened due to the interaction between the stiffer matrix and the relatively softer graphite inclusions. Upon reaching 6% strain (Figure 10c), the stress distribution becomes more heterogeneous, with high-stress zones developing along the periphery of graphite particles and their immediate surroundings. The maximum effective stress values reach approximately 400 MPa, indicating that significant portions of the matrix have transitioned from elastic to plastic deformation. Despite this, the overall structure maintains its integrity, supported by the effective load transfer facilitated by the spherical graphite morphology. At 9% strain (Figure 10d), the stress field shows pronounced gradients near particle–matrix interfaces, highlighting the initiation of damage processes such as microvoid formation and interfacial debonding. However, unlike brittle materials prone to rapid failure, the surrounding matrix continues to support substantial loads, demonstrating the capacity for sustained deformation and energy absorption through plastic flow. As the strain increases to 12% (Figure 10e), the development of macroscopic cracks begins to influence the stress landscape. While stress concentrations decrease in the vicinity of these cracks, other regions maintain high stress levels, illustrating the ability of the material to redistribute loads away from damaged zones. This characteristic enables the material to endure higher strains before ultimate failure. Finally, at 15% strain (Figure 10f), a dominant crack has propagated throughout the RVE, resulting in a large area of reduced stress. Nevertheless, residual high-stress zones persist near the crack tip, suggesting ongoing plastic deformation and energy dissipation mechanisms even in the late stages of loading. This observation underscores the role of progressive damage evolution in determining the final fracture behavior of the material.

The presented stress evolution provides critical information on the underlying mechanics governing the deformation process in ductile cast iron. It highlights the importance of graphite morphology in influencing not only the initial yield but also the subsequent hardening and damage accumulation phases. By mitigating sharp stress concentrations and facilitating effective load redistribution, spheroidal graphite inclusions contribute significantly to enhancing the overall ductility and toughness of the material.

The progression of effective stress within the ferritic matrix of gray cast iron containing flake graphite is illustrated in Figure 11, revealing the distinct mechanical response compared to spheroidal graphite structures. At zero strain (Figure 11a), the RVE exhibits a uniform low-stress state across both matrix and graphite phases, consistent with elastic behavior. As the applied strain increases to 3% (Figure 11b), stress begins to concentrate along the edges of the flake graphite inclusions, particularly at their sharp tips. These regions act as stress raisers, promoting early local yielding in the surrounding matrix. The presence of elongated and irregularly oriented flakes leads to anisotropic stress fields, which are more prone to localized failure than those observed in spheroidal microstructures. At 6% strain (Figure 11c), significant stress gradients develop around the graphite flakes, with peak values exceeding 400 MPa near the sharp ends. This indicates that plastic deformation has initiated in the matrix adjacent to these high-stress zones. However, unlike ductile cast iron, there is limited ability for stress redistribution due to the geometric discontinuities introduced by the flake morphology. By 9% strain (Figure 11d), microcracks nucleate at the graphite-matrix interfaces, especially at the tips of the flakes where triaxial stress states prevail. These cracks propagate rapidly along the plane of the flakes, leading to a sudden drop in local stress levels in affected areas. The onset of damage results in load shedding from the fractured regions, but the remaining intact matrix continues to carry the load. At 12% strain (Figure 11e), multiple microcracks coalesce into larger fissures, forming continuous crack paths that traverse the RVE. The stress field becomes highly fragmented, with isolated high-stress pockets persisting only in undamaged regions. The loss of load-bearing capacity is evident as the overall stress level decreases significantly in the vicinity of the growing fracture zone. Finally, at 15% strain (Figure 11f), a dominant macroscopic crack has fully developed and propagated through the specimen, resulting in complete separation of the material. The stress distribution shows extensive regions of low or zero stress, indicating that the structural integrity has been compromised. The final fracture surface aligns closely with the orientation of the graphite flakes, confirming their role as preferential crack propagation paths.

This sequence highlights the detrimental effect of flake graphite on the mechanical performance of cast iron under tension. The sharp geometry of the flakes induces severe stress concentrations even at low strains, leading to premature crack initiation and rapid failure. In contrast to the gradual damage accumulation seen in ductile iron, the failure mechanism here is dominated by brittle-like crack propagation, despite the presence of a ductile matrix.

A direct comparison of Figure 10 and Figure 11 highlights the profound influence of graphite morphology on stress distribution and failure evolution in cast iron. In ductile cast iron with spheroidal graphite, stress concentrates gradually at particle-matrix interfaces, enabling plastic deformation, strain hardening, and effective load redistribution, which delay fracture and promote ductile behavior. In contrast, gray cast iron with flake graphite exhibits early and intense stress localization at the sharp tips of graphite flakes, acting as potent stress raisers that initiate microcracks at low strains. These cracks propagate rapidly along the flake planes, leading to premature failure with minimal energy absorption. The spheroidal morphology thus promotes a more favorable stress state and progressive damage evolution, while the flake structure creates inherent discontinuities that severely compromise load-bearing capacity and fracture resistance.

## 4. Conclusions

In this study, a finite element method (FEM)-based micromechanical framework is proposed for the performance analysis and mechanical property prediction of cast iron with different graphite morphologies. Three key conclusions are drawn:

(1) The developed FEM model effectively captures the evolution of plastic strain and stress distribution in representative volume elements (RVEs), enabling quantitative insight into the micromechanical mechanisms underlying macroscopic deformation and failure.

(2) Graphite morphology is identified as the dominant factor influencing mechanical behavior: spheroidal graphite promotes ductile deformation through controlled stress concentration and progressive damage accumulation, whereas flake graphite induces severe stress localization and early crack initiation, leading to brittle-like failure.

(3) The simulation results demonstrate that the proposed approach can accurately predict the mechanical response and failure pathways in cast iron, offering a reliable tool for microstructure-sensitive performance assessment and material design optimization.

This work establishes a direct link between microstructural features and mechanical performance, highlighting the potential of FEM-based modeling as a predictive tool for cast iron alloys with diverse graphite morphologies.

## Figures and Tables

**Figure 1 materials-18-05128-f001:**
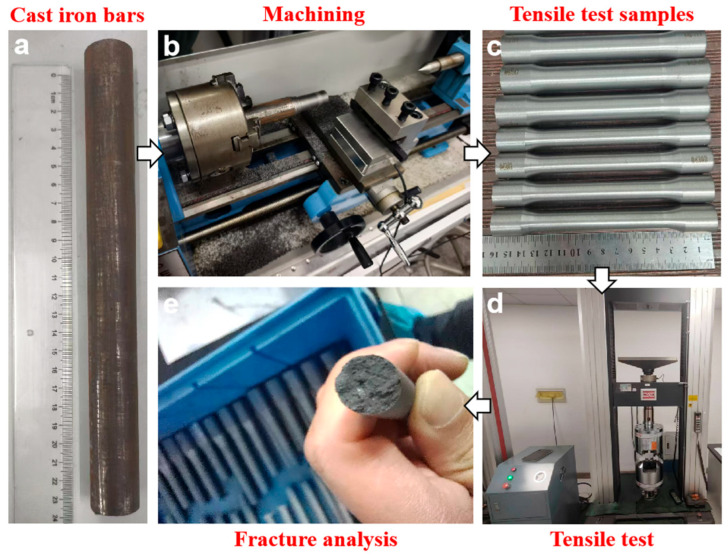
Schematic illustration of the sample preparation and tensile testing process for cast iron.

**Figure 2 materials-18-05128-f002:**
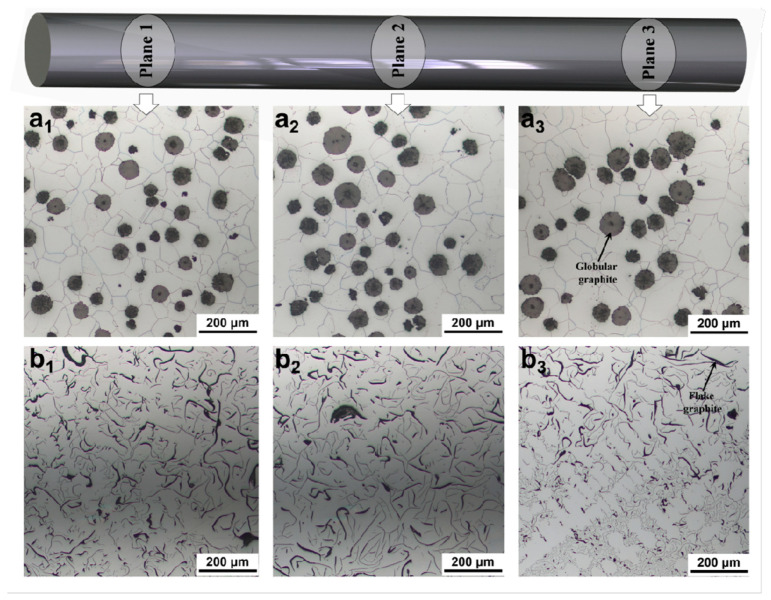
Microstructural characterization of (**a_1_**–**a_3_**) ductile iron and (**b_1_**–**b_3_**) gray iron across three different cross-sectional planes (Plane 1, Plane 2, Plane 3).

**Figure 3 materials-18-05128-f003:**
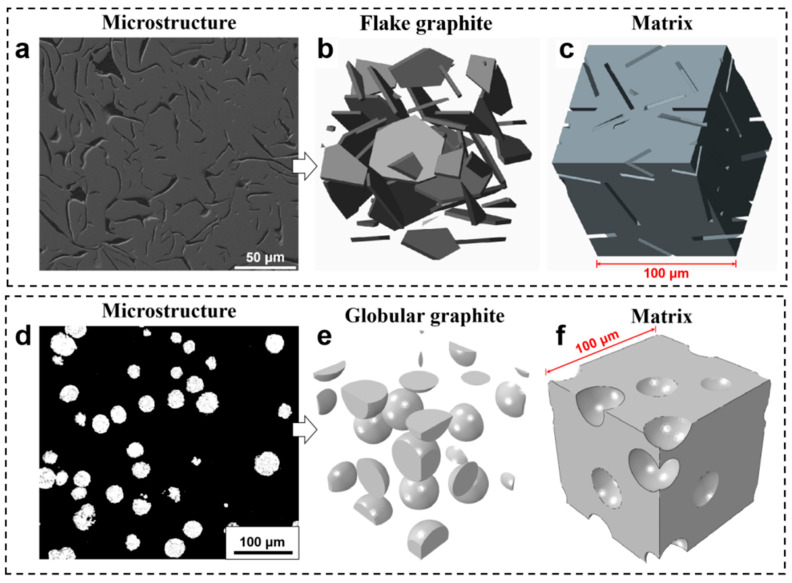
Representative 3D microstructural models of gray iron (**a**–**c**) and ductile iron (**d**–**f**) based on real graphite morphologies.

**Figure 4 materials-18-05128-f004:**
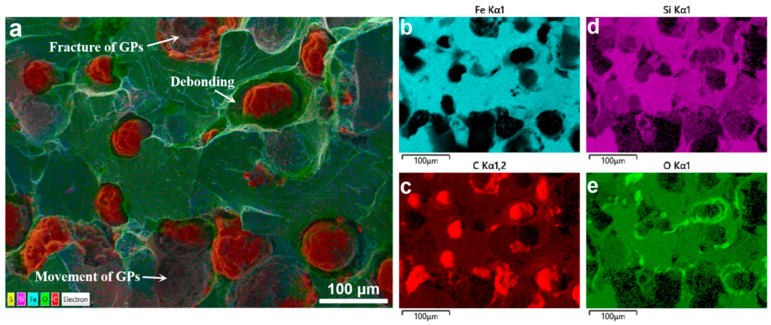
SEM and EDS map scanning of the fracture surface in ductile iron: (**a**) SEM image showing graphite particle fracture, debonding, and displacement; (**b**–**e**) EDS maps of Fe, C, Si, and O distributions.

**Figure 5 materials-18-05128-f005:**
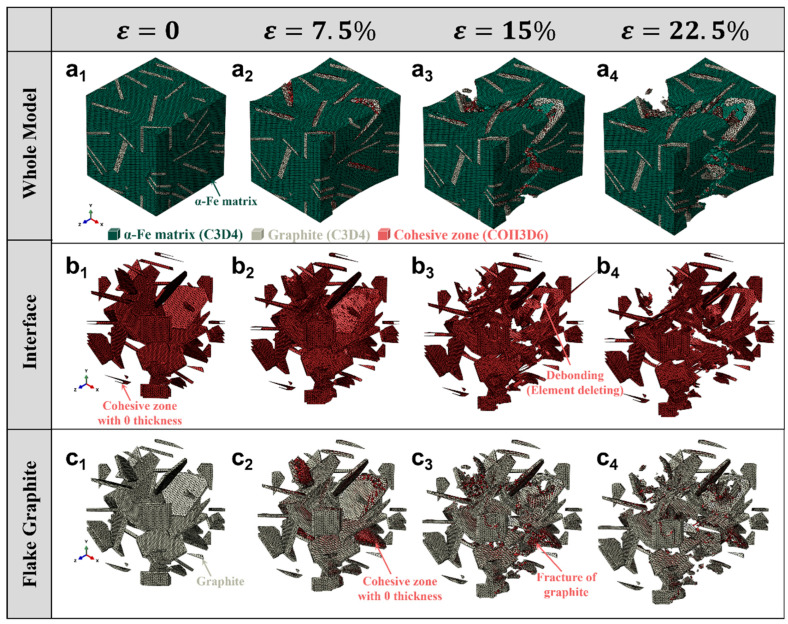
Evolution of damage in gray iron under tensile loading with varying strain *ε*: (**a_1_**–**a_4_**) Overall microstructure showing matrix deformation and crack initiation; (**b_1_**–**b_4_**) Interfacial debonding at the graphite/matrix interface modeled by cohesive zones (COH3D6); (**c_1_**–**c_4_**) Fracture progression within flake graphite particles, with element deletion indicating brittle failure.

**Figure 6 materials-18-05128-f006:**
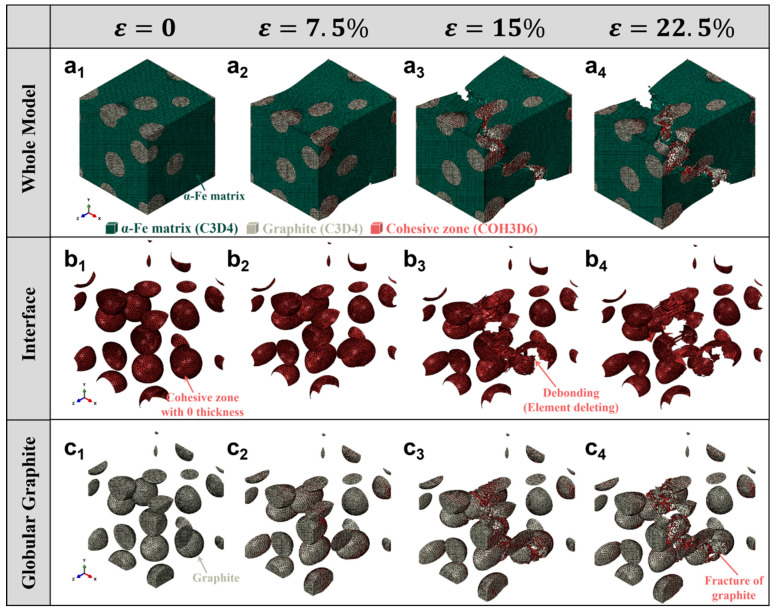
Damage evolution in ductile iron with globular graphite under tensile loading with varying strain: (**a_1_**–**a_4_**) Global deformation and crack propagation in the α-Fe matrix; (**b_1_**–**b_4_**) Interfacial debonding at the graphite/matrix interface modeled by zero-thickness cohesive zones; (**c_1_**–**c_4_**) Fracture initiation and progression within spherical graphite particles, indicating brittle failure.

**Figure 7 materials-18-05128-f007:**
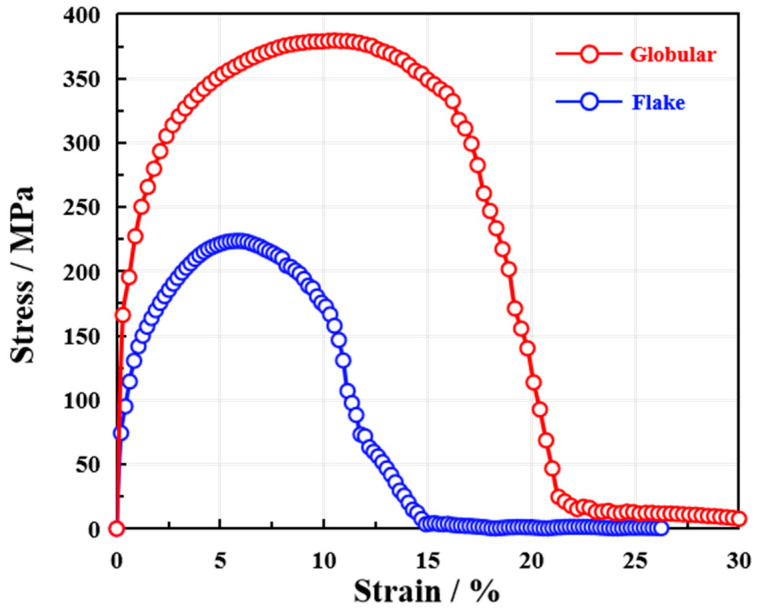
Simulated tensile stress–strain curves for cast iron with globular and flake graphite morphologies.

**Figure 8 materials-18-05128-f008:**
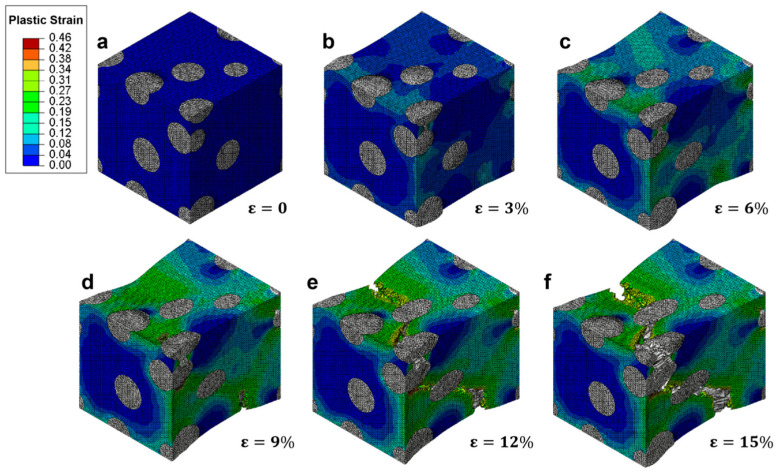
Evolution of plastic strain distribution in a representative volume element (RVE) of ductile cast iron under tensile loading at increasing strain levels: (**a**) ε = 0%, (**b**) ε = 3%, (**c**) ε = 6%, (**d**) ε = 9%, (**e**) ε = 12%, and (**f**) ε = 15%.

**Figure 9 materials-18-05128-f009:**
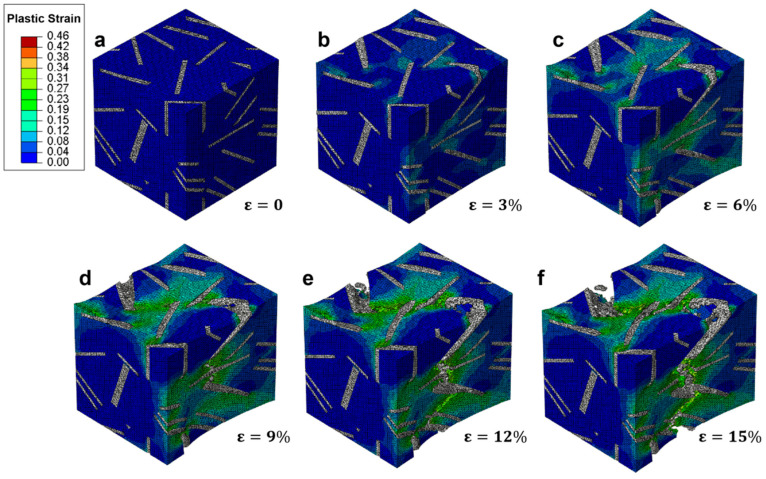
Evolution of plastic strain distribution in a RVE of gray cast iron under tensile loading at increasing strain levels: (**a**) ε = 0%, (**b**) ε = 3%, (**c**) ε = 6%, (**d**) ε = 9%, (**e**) ε = 12%, and (**f**) ε = 15%.

**Figure 10 materials-18-05128-f010:**
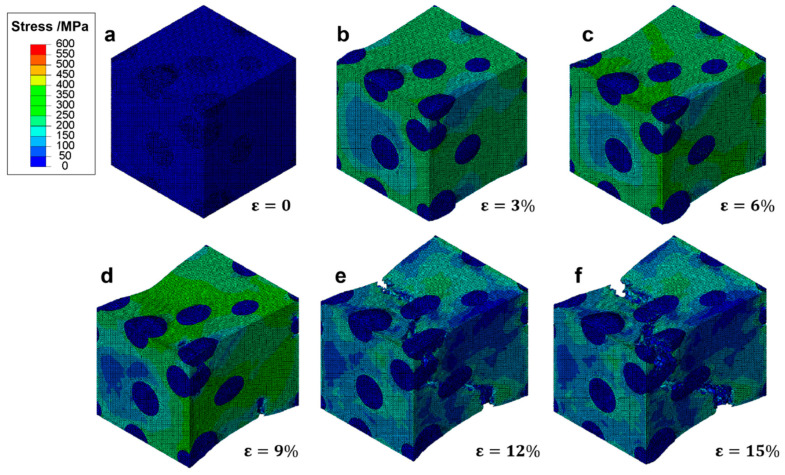
Effective stress distribution in a RVE of ductile cast iron at various strain levels: (**a**) ε = 0%, (**b**) ε = 3%, (**c**) ε = 6%, (**d**) ε = 9%, (**e**) ε = 12%, and (**f**) ε = 15%.

**Figure 11 materials-18-05128-f011:**
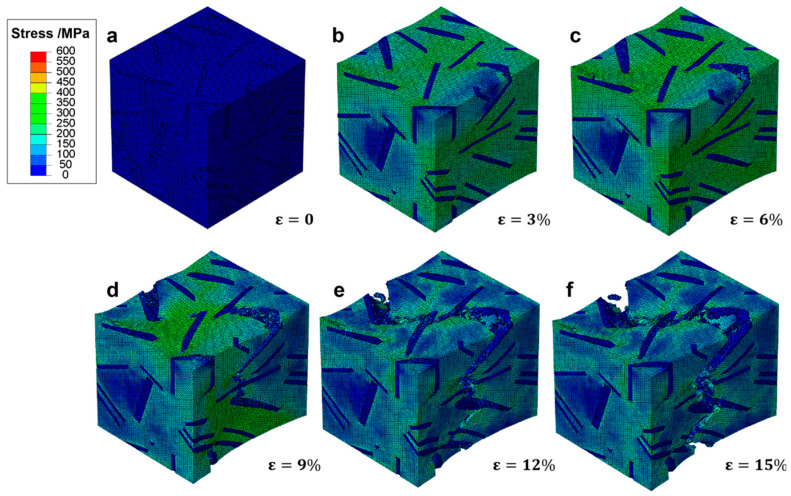
Evolution of effective stress distribution in a RVE of cast iron with flake graphite under tensile loading at increasing strain levels: (**a**) ε = 0%, (**b**) ε = 3%, (**c**) ε = 6%, (**d**) ε = 9%, (**e**) ε = 12%, and (**f**) ε = 15%.

**Table 1 materials-18-05128-t001:** Parameters of the materials used in FEM.

Parameters	Ductile Iron	Gray Iron
Graphite Volume Fraction (Measurement)	19.7%	18.8%
Graphite Volume Fraction (FEM Model Setting)	20%	20%
Average Particle Size (FEM)	-	-
Diameter	25 ± 5 μm	-
Length	-	45 ± 9 μm
Thickness	-	3 μm
Average Nearest-Neighbor Distance	25 μm	25 μm
RVE Dimensions	100 μm × 100 μm × 100 μm	100 μm × 100 μm × 100 μm
Average Mesh Size	~2 μm	~2 μm

## Data Availability

The original contributions presented in this study are included in the article. Further inquiries can be directed to the corresponding author.

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
