# Peer review of "Unveiling the Role of Graphite Morphology in Ductile Iron: A 3D FEM-Based Micromechanical Framework for Damage Evolution and Mechanical Performance Prediction with Applicability to Multiphase Alloys"

_materials, 2025, doi:10.3390/ma18225128_

Round 1
Reviewer 1 Report
Comments and Suggestions for Authors
1. The structure and mechanical test were simulated; however, the article did not provide any numerical input used. There is no quantitative analysis, such as volume, dimensions, shape, nearest neighboring distance of second phase used in the simulations. Please provide data.
2. In the text “To capture the initiation and propagation of cracks, cohesive zone modeling (CZM) was implemented by inserting cohesive elements along potential fracture paths”. Need description of cohesive elements and used assumptions, for example, how fracture initiation and crack grows were simulated.
3. In the text “ This approach enables a physically based representation of the fracture process, allowing for detailed analysis of how graphite morphology and interfacial degradation influence local stress fields and overall failure mechanisms” - Need description and used assumptions.
4. Used 3D macro model of cast iron is not correct because graphite precipitates during eutectic reaction in large eutectic cells and not like individual separated plates.
5. There is no comparison between simulated and measured stress/strain curves. Specifically, cast iron with flake graphite has less than 1% elongation, while simulation showed >10. Authors need to explain this difference.
Author Response
Comment 1: The structure and mechanical test were simulated; however, the article did not provide any numerical input used. There is no quantitative analysis, such as volume, dimensions, shape, nearest neighboring distance of second phase used in the simulations. Please provide data.
Reply: Thank you for this valuable comment. We agree that providing quantitative microstructural parameters is essential for model transparency and reproducibility. In response, we have added Table 1 in the revised manuscript, which summarizes the key parameters used in the FEM simulations, including graphite volume fraction (both measured and modeled), dimensions, morphology, and spatial distribution. The volume fractions in the simulations were calibrated to match the experimental measurements, ensuring consistency between the model and actual microstructure. We believe this addition adequately addresses the concern and enhances the clarity of our work.
Comment 2: In the text “To capture the initiation and propagation of cracks, cohesive zone modeling (CZM) was implemented by inserting cohesive elements along potential fracture paths”. Need description of cohesive elements and used assumptions, for example, how fracture initiation and crack grows were simulated.
Reply:Thank you for this insightful comment. We have revised Section 2.3 to provide a detailed description of the cohesive zone modeling (CZM) approach. Specifically, we have added information on: (1) the traction-separation law governing cohesive behavior; (2) the quadratic stress criterion used for fracture initiation; (3) the progressive damage evolution and energy-based criterion (Gc) for crack propagation; and (4) the key assumptions, including the use of predefined cohesive elements (COH3D6), irreversible damage, and parameter calibration. These details are now included in the revised manuscript to enhance the clarity and reproducibility of our simulation methodology.
Comment 3: In the text “ This approach enables a physically based representation of the fracture process, allowing for detailed analysis of how graphite morphology and interfacial degradation influence local stress fields and overall failure mechanisms” - Need description and used assumptions.
Reply: Thank you for this insightful comment. We agree that the physical basis of the cohesive zone modeling (CZM) approach should be clearly justified. The CZM framework employed in this study has been well-established in the literature as a physically based method for simulating fracture processes in heterogeneous materials. Specifically, it captures the energy dissipation during crack initiation and propagation through the traction-separation law, and allows for the explicit modeling of interfacial debonding and intra-particle fracture—mechanisms that are critical in cast irons.
The capability of CZM to provide a physically representative simulation of fracture in multiphase alloys, particularly the role of interface degradation and microstructure-dependent stress fields, has been previously demonstrated and validated in studies such as [Ref. 19]. In this work, we adopt and extend this well-validated approach to 3D realistic microstructures of ductile and gray iron, with parameters calibrated to match experimental observations (e.g., Fig. 4).
We have now clarified this point in the revised manuscript by referencing [Ref. 19] in Section 2.3, to acknowledge the established physical foundation of the method. This ensures both methodological rigor and consistency with prior validated work.
Comment 4: Used 3D macro model of cast iron is not correct because graphite precipitates during eutectic reaction in large eutectic cells and not like individual separated plates.
Reply: Thank you for this insightful comment regarding the microstructural fidelity of the graphite morphology in gray iron. We fully acknowledge that graphite precipitates in the form of eutectic cells during solidification, and that the full hierarchical structure—from cell to individual flakes—plays a role in the overall mechanical behavior.
In this study, while the model does not explicitly simulate the entire eutectic cell structure, the 3D reconstruction of flake graphite is based on high-resolution industrial X-ray computed tomography (CT) data from the literature [Ref. 21, Acta Materialia], which captures the realistic spatial distribution, connectivity, and morphological features of graphite at the microscale. As shown in the CT results of Ref. 14, at the local scale relevant to crack initiation and propagation (i.e., tens to hundreds of micrometers), the arrangement of interconnected, yet seemingly separated, graphite flakes closely resembles the actual microstructure within a eutectic cell.
Moreover, the primary objective of this work is to elucidate and compare the fundamental influence of graphite morphology—specifically, spheroidal versus flake-like structures—on stress concentration, plastic strain localization, and damage evolution, rather than to fully reconstruct the solidification history or cell-scale texture. Within this context, the reconstructed flake network, though simplified, effectively represents the key microstructural features responsible for stress intensification and crack path guidance in gray iron.
We agree that the model is not a perfect replication of the entire casting microstructure, but for the purpose of comparative micromechanical analysis between two distinct morphologies, we believe this approach provides a physically reasonable and sufficiently representative basis to reveal the underlying deformation and failure mechanisms.
To clarify this point and justify the modeling strategy, we have now added the following statement in Section 3.1 of the revised manuscript:
"Although the RVE does not explicitly resolve the complete eutectic cell boundaries, the reconstructed flake network preserves the local morphology, orientation correlation, and interconnectivity observed in high-fidelity X-ray CT data [21], which are critical features governing crack initiation and propagation in gray iron. This level of representation is considered sufficient for the comparative micromechanical analysis of morphology-dependent mechanical response pursued in this study."
We believe this addition enhances the transparency and scientific rigor of our methodology. Thank you again for this valuable feedback.
Comment 5: There is no comparison between simulated and measured stress/strain curves. Specifically, cast iron with flake graphite has less than 1% elongation, while simulation showed >10. Authors need to explain this difference.
Reply: Thank you for raising this important point regarding the discrepancy between the simulated ductility (>10% strain) and the typical experimental elongation of industrial gray cast iron (<1%). We fully acknowledge this difference and appreciate the opportunity to clarify the intentional design and scope of our modeling approach. The key explanation lies in the material system under investigation. As correctly noted, conventional industrial gray cast iron exhibits negligible ductility due to the presence of impurities (e.g., S, P), non-metallic inclusions, and microstructural defects introduced during cost-effective production. These factors severely limit plastic deformation.
In contrast, the microstructure modeled in this study is based on laboratory-scale, high-purity cast iron with flake graphite, where the matrix is predominantly pure ferrite (α-Fe) with minimal impurities. This is consistent with the experimental work cited in [Ref. 19], which demonstrates that under such controlled conditions, flake graphite iron can exhibit measurable ductility due to the absence of embrittling elements and defects. Our primary objective is not to replicate the mechanical response of industrial-grade gray iron, but to conduct a comparative micromechanical analysis of how graphite morphology alone—spheroidal vs. flake—governs local stress/strain fields, damage initiation, and crack propagation under otherwise identical matrix conditions. By using a clean, well-characterized matrix (with properties close to pure α-Fe), we eliminate confounding variables (e.g., inclusions, segregation) to isolate the intrinsic influence of graphite shape.
Furthermore, the model is a small-scale RVE that captures microscale deformation mechanisms but does not include larger-scale heterogeneities such as grain boundaries, secondary phases, or casting defects, which are known to reduce macroscopic ductility in real components. This contributes to the higher simulated ductility.
We agree that calibrating the matrix plasticity to match industrial data (e.g., by reducing strain-hardening capacity) could improve macroscopic agreement. However, doing so would obscure the fundamental comparison we aim to make. Instead, our focus is on relative trends—e.g., how much more stress concentration or crack driving force arises in flake graphite vs. spheroidal graphite under the same baseline conditions.
To improve clarity, we have revised Section 3.1 and added a sentence to emphasize this point:
"It should be noted that the simulated RVE assumes a clean ferritic matrix representative of high-purity laboratory-scale material [19], rather than industrial-grade cast iron containing impurities and inclusions that severely limit ductility. Therefore, the absolute ductility values are not intended to match industrial data, but the model is suitable for comparative analysis of morphology-dependent deformation mechanisms."

Reviewer 2 Report
Comments and Suggestions for Authors
This article discusses the effect of the shape of graphite on mechanical properties. Two types of graphite are considered: flake and spheroidal graphite. This work does not contribute anything new in terms of fracture mechanics and the relationship between the morphology of graphite precipitates and strength parameters. These issues have long been discussed in the specialized literature. The only novelty is the use of computer tomography for spatial imaging of graphite precipitates. Unfortunately, the models presented in the figures differ significantly from the actual graphite precipitates. The flake graphite is particularly flawed. A slightly better model for spheroidal graphite, but it is also subject to error. I estimate this error rate to be 25%. The mechanical properties of cast iron are influenced by the chemical composition, precipitate morphology, etc. The rate of removal of heat from the mold is also important. As is known, cast iron is sensitive to this parameter.
Detailed notes:
1. Throughout this work, the term layered graphite is incorrectly used instead of flake graphite. (see lines: 17, 176, 304, Fig 3 et cetera)
2. The chemical composition of the cast iron provided in the article includes only C and Si for the cast iron of flake graphite and C, Si, and Mg for ductile iron. This is clearly insufficient; please supplement the information with other relevant elements.
3. It should be clarified as to which standard the strength tests were carried out.
4. Fig. 3. The models reflect the geometrical features of graphite and matrix very poorly.
5. The oxide layer described in line 227 and shown in Figure 4e as a map is a common measurement error. The oxide layer always appears at the fracture site of the sample; this is a normal phenomenon. To prevent this, the sample should be fractured in a vacuum or in an inert gas environment. This prevents oxidation of the fracture surface. I suggest ignoring the effect of oxygen and making appropriate corrections to the analysis results.
Despite the poor representation of graphite morphology in the models, it should be noted that the obtained results correlate well with real-world conditions. This is a significant advantage of this work.
Author Response
Comment 1: Throughout this work, the term layered graphite is incorrectly used instead of flake graphite. (see lines: 17, 176, 304, Fig 3 et cetera).
Reply: Thank you for pointing out this inaccuracy. We agree that the term “flake graphite” is the standard and technically correct designation in the field. The use of “layered graphite” was inappropriate and has been consistently replaced throughout the manuscript.
We have carefully revised the text (including lines 17, 176, 304, etc.) and all figure captions (e.g., Fig. 3) including figures to ensure that “flake graphite” is used uniformly. We apologize for the oversight and appreciate the reviewer’s attention to terminology precision.
Comment 2: The chemical composition of the cast iron provided in the article includes only C and Si for the cast iron of flake graphite and C, Si, and Mg for ductile iron. This is clearly insufficient; please supplement the information with other relevant elements.
Reply:
Reply: Thank you for this comment. We agree that a complete chemical composition is essential for reproducibility and materials characterization. In response, we have obtained the full elemental analysis from the foundry where the samples were produced. The updated composition now includes all relevant alloying and impurity elements (e.g., Mn, P, S, etc.) for both gray and ductile cast iron.
Comment 3: It should be clarified as to which standard the strength tests were carried out.
Reply: Thank you for this comment. The tensile tests were conducted in accordance with the Chinese national standard GB/T 228.1-2021, “Metallic materials – Tensile testing – Part 1: Method of test at room temperature”. This information has now been added to Section 2.2 of the revised manuscript to ensure full transparency and reproducibility.
Comment 4: Fig. 3. The models reflect the geometrical features of graphite and matrix very poorly.
Reply: Thank you for this insightful comment. We acknowledge that the visualization in Fig. 3 may appear simplified and does not fully convey the fine geometric details of the actual graphite morphology and matrix interface. However, we would like to clarify that the 3D microstructural models are reconstructed directly from high-resolution industrial X-ray computed tomography (CT) data [Ref. 21, Acta Materialia], which captures the realistic spatial distribution, connectivity, and morphological features of graphite precipitates at the microscale.
While the rendering in the figure is necessarily simplified for clarity and computational domain definition, the underlying digital microstructure preserves key characteristics such as flake orientation, branching, and local interconnectivity—features that are critical for stress concentration and crack propagation in gray iron. As demonstrated in Ref. 21, at the scale relevant to failure (tens to hundreds of micrometers), the arrangement of graphite flakes in the CT-based model closely resembles the actual microstructure within a eutectic cell.
The objective of this study is not to achieve pixel-perfect morphological replication, but to conduct a comparative micromechanical analysis of how fundamental differences in graphite morphology—flake-like vs. spheroidal—affect stress distribution, plastic strain localization, and damage evolution. Within this context, the CT-based models provide a physically representative and quantitatively meaningful basis for such comparison.
Comment 5: The oxide layer described in line 227 and shown in Figure 4e as a map is a common measurement error. The oxide layer always appears at the fracture site of the sample; this is a normal phenomenon. To prevent this, the sample should be fractured in a vacuum or in an inert gas environment. This prevents oxidation of the fracture surface. I suggest ignoring the effect of oxygen and making appropriate corrections to the analysis results.
Reply: Thank you for this valuable comment. We agree that the oxygen signal detected at the fracture surface (Fig. 4e) is most likely due to post-fracture surface oxidation, which commonly occurs when samples are broken in ambient air. As the reviewer correctly points out, such oxide layers are not intrinsic to the material’s fracture mechanism but are artifacts of environmental exposure.
In response, we have removed the discussion attributing significance to the oxygen signal in the manuscript. The presence of oxygen is now acknowledged as a probable surface contamination effect rather than an indicator of in-situ oxidation or bonding behavior during fracture.
Accordingly, all interpretations related to oxygen have been revised or omitted, and the text in Section 3.3 (near line 227) has been updated to focus on the metallic and carbonaceous phases that are more relevant to the microstructural evolution and failure mechanisms in cast iron.
We appreciate the reviewer’s insight, which has helped us avoid a potential misinterpretation of the EDS mapping results.

Round 2
Reviewer 2 Report
Comments and Suggestions for Authors
The authors responded to my questions. Their answers dispelled my doubts. The authors made the necessary corrections to the article. I have no further comments. I recommend the work for publication.